# Network propagation of rare variants in Alzheimer's disease reveals tissue-specific hub genes and communities

**Marzia Antonella Scelsi**[1]*, **Valerio Napolioni**[2¤], **Michael D. Greicius**[2],
**Andre Altmann**[1], **for the Alzheimer's Disease Neuroimaging Initiative (ADNI) and the
Alzheimer's Disease Sequencing Project (ADSP)**[¶]

**1** Centre for Medical Image Computing, Department of Medical Physics and Biomedical Engineering, University College London, London, United Kingdom, **2** Functional Imaging in Neuropsychiatric Disorders (FIND) Lab, Department of Neurology and Neurological Sciences, Stanford University School of Medicine, Stanford, California, United States of America

¤ Current address: Genomic and Molecular Epidemiology (GAME) Lab, School of Biosciences and Veterinary Medicine, University of Camerino, Camerino, Italy
¶ Membership for the Alzheimer's Disease Neuroimaging Initiative (ADNI) and the Alzheimer's Disease Sequencing Project (ADSP) is listed in the Supplementary information S2 Text.
* marzia.scelsi.15@ucl.ac.uk

**Data Availability Statement:** Data underlying the findings are available to qualified researchers. ADSP WES data files are available with restrictions

## Abstract

State-of-the-art rare variant association testing methods aggregate the contribution of rare variants in biologically relevant genomic regions to boost statistical power. However, testing single genes separately does not consider the complex interaction landscape of genes, nor the downstream effects of non-synonymous variants on protein structure and function. Here we present the NETwork Propagation-based Assessment of Genetic Events (NETPAGE), an integrative approach aimed at investigating the biological pathways through which rare variation results in complex disease phenotypes. We applied NETPAGE to sporadic, late-onset Alzheimer's disease (AD), using whole-genome sequencing from the AD Neuroimaging Initiative (ADNI) cohort, as well as whole-exome sequencing from the AD Sequencing Project (ADSP). NETPAGE is based on network propagation, a framework that models information flow on a graph and simulates the percolation of genetic variation through tissue-specific gene interaction networks. The result of network propagation is a set of smoothed gene scores that can be tested for association with disease status through sparse regression. The application of NETPAGE to AD enabled the identification of a set of connected genes whose smoothed variation profile was robustly associated to case-control status, based on gene interactions in the hippocampus. Additionally, smoothed scores significantly correlated with risk of conversion to AD in Mild Cognitive Impairment (MCI) subjects. Lastly, we investigated tissue-specific transcriptional dysregulation of the core genes in two independent RNA-seq datasets, as well as significant enrichments in terms of gene sets with known connections to AD. We present a framework that enables enhanced genetic association testing for a wide range of traits, diseases, and sample sizes.

from dbGaP (accession ID: phs000572.v7.p4), whereas ADNI WGS data files are available without restrictions from ida.loni.usc.edu.

**Funding:** MAS acknowledges financial support by the EPSRC-funded UCL Centre for Doctoral Training in Medical Imaging (EP/L016478/1). MDG was supported by the NIH (P50 AG047366). AA holds an MRC eMedLab Medical Bioinformatics Career Development Fellowship. This work was supported by the Medical Research Council [grant number MR/L016311/1]. The funders had no role in study design, data collection and analysis, decision to publish, or preparation of the manuscript.

**Competing interests:** The authors have declared that no competing interests exist.

## Author summary

In the biomedical field there is ever increasing availability of data from sequencing-based methods, such as whole-genome or whole-exome sequencing, that can greatly help elucidate the role of rare genetic variants in the aetiology of common diseases. However, state-of-the-art rare variant association methods are vastly underpowered in small to medium-sized studies and therefore novel methodologies are needed to leverage these datasets while integrating information from different genomic sources. To this end we present NETPAGE, a gene-based association testing method that models how the effect of rare deleterious variants spreads over gene interaction networks. NETPAGE is robust and flexible, and can be applied to different diseases, sample sizes, and types of traits (binary or continuous). We demonstrate the successful application of NETPAGE to two Alzheimer's disease cohorts of different sizes and sequencing methods, identifying connected hub genes and communities underlying biological processes and pathways involved in Alzheimer's and other neurodegenerative diseases, and that could be considered as potential drug targets.

## Introduction

The advent of next generation sequencing (NGS) has drastically changed the genetic landscape of both complex and Mendelian traits, widening the range of approaches to investigate the genetic bases of human phenotypes. Using SNP-array genotyping technologies, large scale studies involving 100,000s of participants are focusing on common variants in order to identify loci associated with complex traits [1]. However, owing to natural selection, common variants typically do not impart major risk for disease [2], with rare exceptions of variants such as the ε4 allele of *APOE* in Alzheimer's disease (AD) [3,4]. Loci identified in genome wide association studies (GWAS) are often flagging the existence of haplotypes harbouring rare variants with strong disease effects and therefore have to be followed up by fine-mapping to identify the true causal variants.

NGS, by contrast, is most effectively used when focusing on rare genetic variants, private mutations or structural genome changes. These types of rare variants have the possibility to exercise a large effect on disease risk and often show a Mendelian inheritance pattern, thus being at the core of familial forms of many disorders; prominent examples are rare variants in *APP*, *PSEN1*, and *PSEN2* in familial AD [5] or variants in *MAPT*, *GRN* and *C9orf72* in frontotemporal dementia (FTD) [6]. The decreasing costs for NGS have enabled the establishment of large whole genome (WGS) or whole exome (WES) sequencing studies such as the AD Sequencing Project (ADSP) [7]; still the largest studies are two orders of magnitude smaller than the largest GWAS (i.e., 10,000s vs a million participants). Moreover, by definition, rare variants are not frequent in the population and it is unlikely for the same rare variant to be shared by many subjects in a study with limited sample size. Therefore, the resulting data matrix of subjects × rare variants is sparse. This drawback is often addressed with alternative study designs such as "extreme phenotyping", based on the assumption that rare variants accumulate in the extreme tails of the phenotype distribution.

Low sample size and sparsity pose a problem for analysing these types of genetic data: two determinants of the statistical power for classical association studies are sample size and allele frequency, leading to low statistical power in rare variant association analyses when relying on the typical single-variant GWAS methodology. This problem has motivated the development of advanced statistical methods, reviewed elsewhere [8]. The most straightforward approach is

the gene-wise burden test, i.e., for each gene the number of rare nonsynonymous variants is counted per participant and the gene burden is compared between cases and controls. The underlying assumption, in order to convert a sparse matrix into a non-sparse one, is that affected individuals tend to carry mutated versions of key genes, while healthy individuals do not. This assumption has been proven to be correct in the case of HDL cholesterol levels [9]. This basic model has been superseded by methods based on the Sequence Kernel Association Test (SKAT) [10], which performs a variance-component test: this assumes that the genetic effect for a given variant (regression coefficient for the SNP in a mixed model) follows a distribution with mean 0 and variance $w_j^2 \tau$, and then tests the null hypothesis $H_0: \tau = 0$. One key advantage of this test is that the rare variants within the gene do not have to influence the disease risk in the same direction. SKAT has been successfully used to study rare variants in a number of diseases such as AD [11], schizophrenia [12], and health and disease more generally [13].

Still, SKAT and its extensions focus on predefined genetic regions, such as genes or even pathways. Recent observations, however, suggest that the underlying nature of genetic effects is more complicated. For instance, the recently proposed omnigenic model for complex traits [14] postulates that genetic variation percolates through gene interaction networks and that, due to the small-world property of such networks, any gene is only a few steps away from the "core" genes with specific roles in disease etiology, such that the effect of variation flows from peripheral genes to "core" genes. In fact, an exonic, deleterious variant within a gene by definition will lead to changes in the resulting protein (e.g., amino acid substitution, premature truncation of transcription, loss of a stop codon). These changes will in turn affect the protein's downstream interactions within the cell environment. Ultimately this may disrupt one or more molecular pathways through a chain reaction (or domino effect). This small-world property implies that variants in different, unrelated genes and different patients can "converge" and exert their effect on the same core gene (Fig 1). Furthermore, complex traits are mediated through multiple tissues or cell types and, consequently, the quantitative effect of genetic variation will vary across tissues owing to differences in gene and protein interaction networks.

In this work we propose a gene-based test for rare variation whose rationale resembles the omnigenic assumption. In particular, we leverage the network propagation approach, which models a diffusion process or information flow on a graph structure, in order to delineate the percolation effect of genetic variation through gene networks. Additionally, we embed tissue

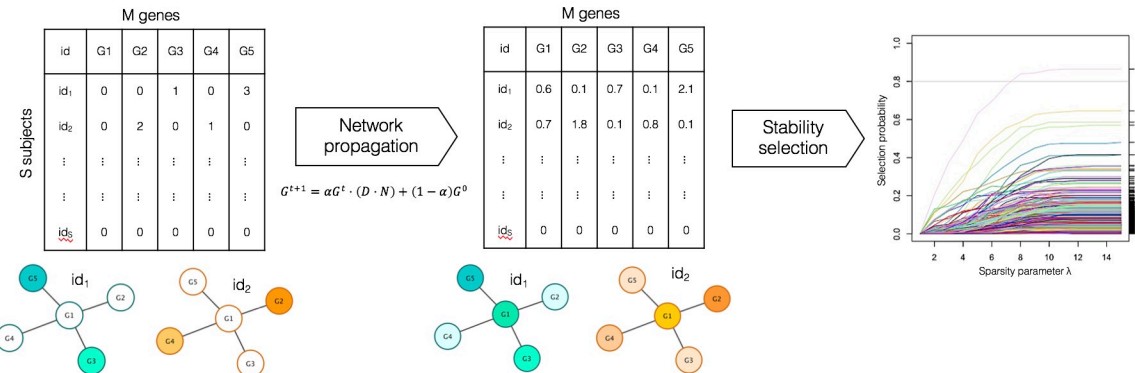

**Fig 1. NETPAGE combines network propagation with sparse regression to follow the pathways by which rare variants percolate through gene interaction networks.** In the network propagation process, the flow of information through the network is controlled by the diffusion length α. The matrix G in the formula represents the variation burden at iterations *t* and *t+1*; while D·N is a degree-normalised adjacency matrix of the gene interaction network (see Methods). The result of network propagation is a set of smoothed gene scores tested for association with disease status through sparse regression and stability selection.

specificity in our framework by using tissue-specific gene-interaction networks obtained from Greene *et al.* [15]. Network propagation is an established approach that has enabled methodological advances as well as important findings in many research fields. In particular, it has been defined as an "universal amplifier of genetic associations" [16]. In brief, network propagation has been successfully applied to various bioinformatics tasks, such as the prioritization of disease-associated genes based on gene interactions and disease similarities [17] or based on a modified version of Google's PageRank algorithm [18], and the stratification of tumour subtypes based on somatic mutation signatures [19]. A more comprehensive review of applications of network propagation can be found elsewhere [16]. Notably, an early attempt at gene prioritisation based on tissue-specific interaction networks dates back to 2012 [20]. However, the networks developed for this work leveraged the tissue specificity of gene expression only, whereas Greene *et al.* [15] integrated a much wider variety of genomic data types, comprising gene co-expression, transcription factor regulation, protein interaction, chemical and genetic perturbations, and microRNA target profiles.

Here we present the NETwork Propagation-based Assessment of Genetic Events (NET-PAGE), an integrative approach that combines network propagation with sparse regression in order to investigate the biological pathways through which genetic variation affects tissue function and results in complex disease phenotypes. Our approach is highly generalisable and enables enhanced genetic association testing for a wide range of complex traits (binary and continuous), diseases, and sample sizes. As a specific proof of concept, we applied NETPAGE to study rare genetic variation in a small whole genome sequencing (WGS) dataset focusing on sporadic late-onset AD obtained from the AD Neuroimaging Initiative (ADNI), as well as a medium-sized whole exome sequencing (WES) dataset from the AD Sequencing Project (ADSP).

## Results

### Method overview

NETPAGE combines network propagation with sparse regression to identify genes robustly associated with a phenotype (Fig 1). In this work we focused on rare, exonic, deleterious Single Nucleotide Variants (SNVs) from WGS or WES; different criteria for the selection of rare deleterious SNV were utilised, to assess their impact on the association testing results (see Methods for details). SNVs were projected onto a gene interaction network. We compared the hippocampus network derived by Greene et al. [15] with the whole, non-tissue-specific human interactome available in STRING [21]. Network propagation is then used to model the propagation of the effects of rare variants through the network. Network propagation results in a set of gene-wise, continuous "smoothed" scores that are then tested for robust association with a disease phenotype through sparse regression (LASSO [22]) and stability selection [23]. NET-PAGE is not intended as a classification tool, hence we did not investigate its prediction performances in the classic machine-learning sense.

### Network propagation: simulations

In order to explore the effect of different parameters in the network propagation algorithm we conducted a series of simulations to map out reasonable default parameters for the algorithm. We chose to implement a simulation framework for this task because the standard approach of optimizing cross-validated prediction performance was not informative in this setting due to the narrow range of performance values. The key parameters of network propagation are the distance a SNV signal is allowed to travel through the network (termed diffusion length α from now on), and the percentage of top edges to be retained in the network in a procedure

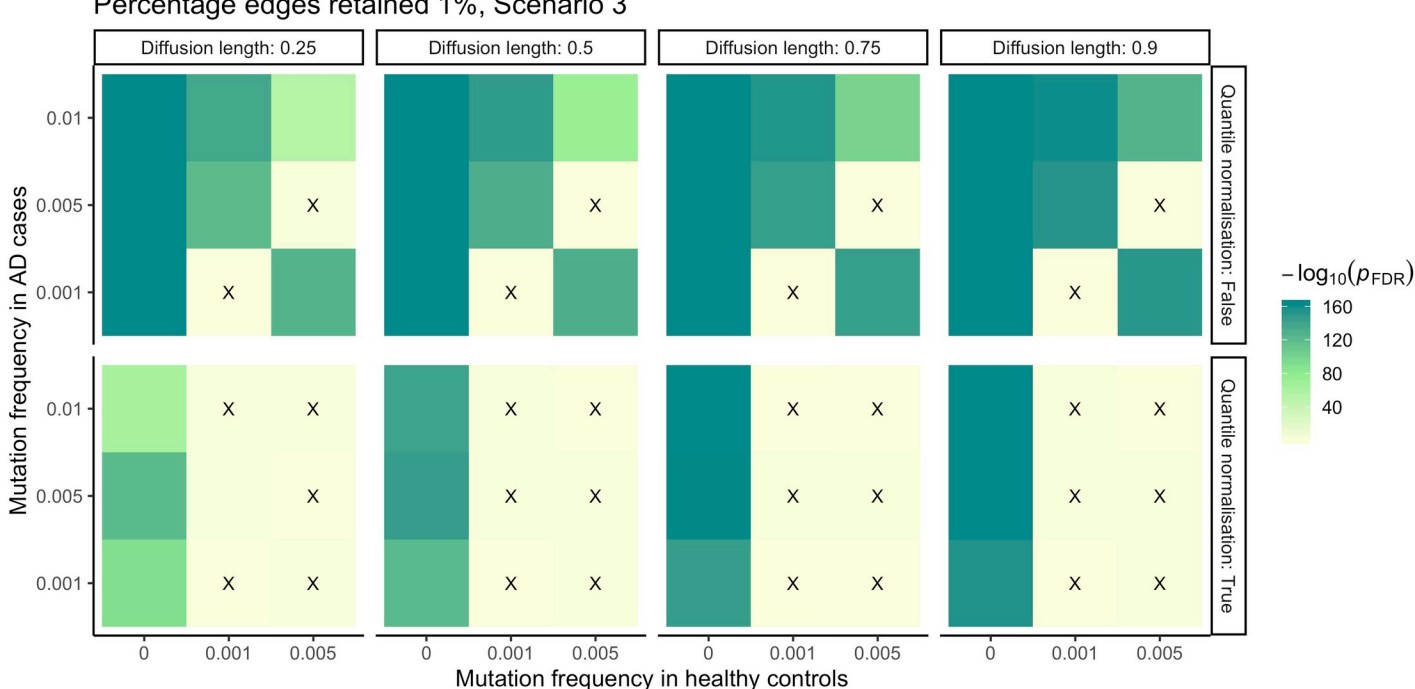

**Fig 2. A selection of simulation results.** The x and y axis represent the SNV frequencies in controls and cases respectively. The faceting allows to visualise the effect of other parameters (diffusion length, quantile normalisation). The colour-coding indicates the statistical significance of the difference in the hub gene smoothed score between controls and cases, in units of -log10 p-value (Bonferroni corrected). We investigated three mutation scenarios: scenario 1, only first neighbours mutated; scenario 2, only second neighbours mutated; scenario 3, both first and second neighbours mutated; the target gene is always unmutated. Cells marked with a black cross indicate parameter combinations where the smooth score of the target gene was not significantly different between cases and controls. This set of tile plots shows the effect of varying diffusion length quantile normalisation after network propagation with top 1% edges retained in scenario 3.

called binarization (see Methods). We generated synthetic datasets, varying parameters of SNV frequency in cases and controls, and applied network propagation using varying parameter settings of the algorithm. In these simulations we focus on an un-mutated target gene and vary the SNV frequency in the gene's neighborhood differentially between cases and controls. Next, we test how well the smoothed score obtained through network propagation in the target gene separates cases from controls. Further details can be found in S1 Text. A summary of results from the investigation of the parameter space for network propagation on simulated data can be found in Figs 2 and S1. We observed (S1B Fig) that variation in both diffusion length α and percentage of edges retained influences the difference in a hub gene's smoothed score between cases and controls only marginally. Therefore we adopted a parsimonious approach and selected as default parameters for real data applications a mid-range α value of 0.5 and a top edge percentage of 1%. As expected, for α = 0 no propagation occurs. We also observed (S1A Fig) a detrimental effect on the smoothed scores (loss of statistical significance) when quantile-normalisation, which is used in Hofree et al. [19], is applied at the end of network propagation, therefore we decided to exclude this step from our pipeline.

## NETPAGE: application to AD sequencing data

After verifying that the network propagation step alone works as expected, we applied the complete NETPAGE framework, with the full-scale hippocampus gene interaction network, to two independent, real-world NGS datasets: one with a small sample size (ADNI), and another with a medium sample (ADSP).

**Small sample: ADNI.** We used NETPAGE in ADNI to test smoothed scores for M = 13,310 genes for association with binary disease outcome in N = 439 Caucasian subjects (222 healthy controls [HC], 217 AD). One gene resulted from stability selection on the smoothed scores in ADNI (Fig 3A): *PFAS* (selection probability = 0.85; Table 1). We replicated the selection of *PFAS* when testing the gene burden propagated through the STRING network (selection probability = 0.85; S2 Fig). No genes were selected when running stability selection on the "raw" SNV profile ($\alpha = 0$, no smoothing), nor when the SNV profile was smoothed through either 30 degree-preserving randomised versions of the hippocampus network or a non-brain-related network (umbilical cord; S3 Fig). The investigation of other, non-AD related brain components or structures (neuron, medulla oblongata, diencephalon) resulted in different or no associations (S4 Fig), hence reinforcing the idea that tissue-specificity is an important factor to consider in association testing.

When a burden including only rare deleterious stop-gain, stop-loss and frameshift SNVs was smoothed (through either the hippocampus or the STRING PPI network) and tested for association with diagnosis, no genes were selected (S5 Fig).

**Medium sample: ADSP.** We used NETPAGE in ADSP to test smoothed scores for M = 16,298 genes for association with binary disease outcome in N = 10,186 Caucasian subjects (4601 HC, 5585 AD). Stability selection on the smoothed scores in ADSP identified 29 genes as robustly associated with case-control status (Fig 3B). Selection probabilities for these 29 genes are reported in Table 1. As an example, Fig 3C shows the subgraph of radius two centred on *CAMK2B*, including interesting second neighbours of relevance to AD dementia or other neuro-degenerative disorders. Eighteen genes selected from ADSP were found to be second neighbours of the gene selected from ADNI (*PFAS*) in the hippocampus network. This number is significantly higher than it would be expected by chance ($p < 10^{-4}$; S5 Fig), indicating that NETPAGE-selected genes in ADSP are overrepresented in the neighbourhood of *PFAS*. As common variants identified through GWAS are often tagging loci harbouring rare variants, S1 Table lists selection probabilities for the 21 genes reported in a recent large scale GWAS for AD [24]. In terms of network properties of the selected genes, S3 Table reports the degree of genes selected both in ADNI and ADSP in the hippocampus network. The average degree of the network is 22, therefore all these genes are hubs in the sense that their degree greatly exceeds the average.

## Gene-based rare variant association testing

NETPAGE was compared to SKAT-O [25], a state-of-the-art method for gene- and set-based association testing of rare variants. We therefore grouped 48,834 rare, non-synonymous SNVs from 439 Caucasian ADNI samples to 13,591 genes and performed a gene-based test, controlling for age, sex, number of *APOE* ε4 alleles, years of education, and population structure (see Methods). Additionally, to conduct a fair comparison to SKAT-O, a mass univariate test of association between the smoothed gene scores and case-control status was also carried out via logistic regression for M = 13,310 genes, controlling for the same confounders listed above. After correction for multiple comparisons, no genes were significantly associated with case-control status in either test (SKAT-O, S7A Fig; mass univariate smoothed, S7B Fig).

We also sought to compare SKAT-O and NETPAGE on a dataset where SKAT-O would be sufficiently powered to detect associations [11]. We therefore grouped 270,165 non-synonymous SNVs from 10,186 Caucasian ADSP samples into 16,630 genes and performed a gene-based test, correcting for age, sex, number of APOE4 alleles, and sequencing centre (sequencing platform was not included as covariate because concordant with centre). After correction for multiple comparisons, no genes were significantly associated with case-control status in ADSP (S8 Fig).

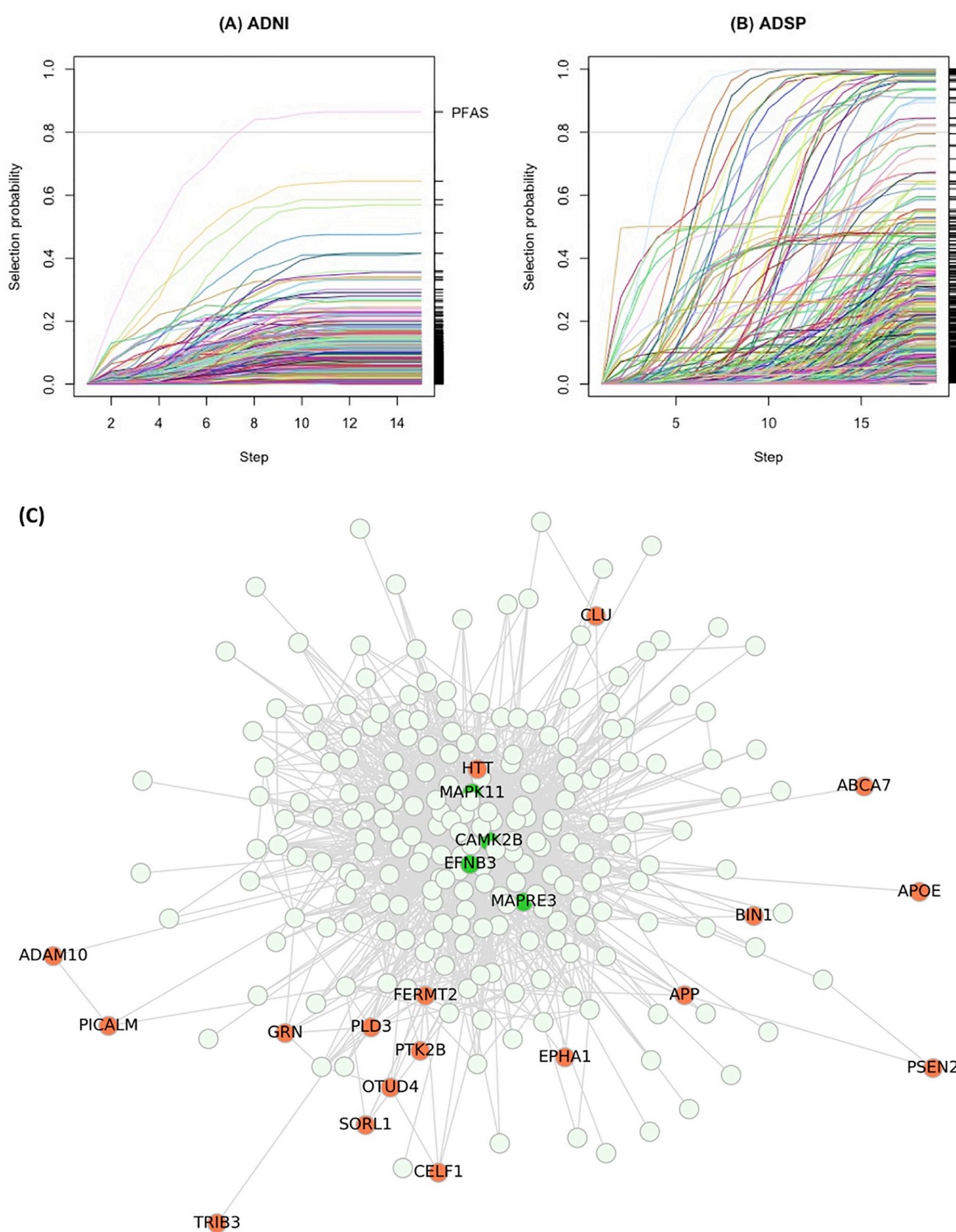

**Fig 3. Stability selection results in ADNI (A) and ADSP (B), for a selection probability cutoff of 0.80.** In these stability path plots, the x axis represents values of the sparsity hyperparameter λ (see Methods, Eq 2) that controls the regularisation required by the LASSO; the y axis represents the selection probability of a gene. Selection probability paths (trajectories) for different genes are represented by different colours. Genes whose trajectories crossed the threshold of 0.80 selection probability were considered as robust predictors of case-control status and followed up in subsequent analyses. Panel (C) shows the ego network of *CAMK2B* (selected in ADSP). Nodes coloured in bright green were identified by NETPAGE; nodes coloured in coral red are genes linked to AD and other neurodegenerative disease in the literature.

## Set-based rare variant association testing

Recent research has shown that the cumulative effect of rare deleterious variation in relation to disease can in some cases be revealed by simply aggregating SNVs into genes and then pathways when running SKAT [26]. To demonstrate the added value of the network propagation

**Table 1. Selection probabilities for the 30 genes whose smoothed score was identified as robustly associated with case-control status in ADNI and ADSP.** Cells shaded in blue contain genes selected in ADNI; cells shaded in green contain genes selected in ADSP. Genes highlighted in bold are second or closer neighbours of *PFAS* in the hippocampus network.

| Gene | Selection probability | Model comparison p-value (Bonferroni corrected) | Gene | Selection probability | Model comparison p-value (Bonferroni corrected) |
|---|---|---|---|---|---|
| **NETPAGE—ADNI** | | | **GNB1** chr 1p36.33 | 1.000 | $1 \times 10^{-29}$ |
| *PFAS* chr 17p13.1 | 0.850 | $2 \times 10^{-4}$ | **HIC2** chr 22q11.21 | 0.990 | $1 \times 10^{-11}$ |
| **NETPAGE—ADSP** | | | **KCNMA1** chr 10q22.3 | 0.980 | $4 \times 10^{-13}$ |
| **ABR** chr 17p13.3 | 0.845 | $4 \times 10^{-16}$ | KLC1 chr 14q32.33 | 1.000 | $1 \times 10^{-23}$ |
| **ADRM1** chr 20q13.33 | 0.965 | $6 \times 10^{-16}$ | MAPK11 chr 22q13.33 | 0.990 | $1 \times 10^{-18}$ |
| *APPBP2* chr 17q23.2 | 0.905 | $4 \times 10^{-22}$ | **MAPRE1** chr 20q11.21 | 0.825 | $1 \times 10^{-16}$ |
| **ARL1** chr 12q23.2 | 0.990 | $3 \times 10^{-22}$ | MAPRE3 chr 2p23.3 | 0.970 | $1 \times 10^{-13}$ |
| *ATXN10* chr 22q13.31 | 0.965 | $7 \times 10^{-17}$ | **MOB4** chr 5p13.1-p12 | 0.895 | $1 \times 10^{-28}$ |
| *CAMK2B* chr 7p13 | 0.960 | $1 \times 10^{-7}$ | **MRPL17** chr 11p15.4 | 0.910 | $1 \times 10^{-24}$ |
| Gene | Selection probability | Model comparison p-value (Bonferroni corrected) | Gene | Selection probability | Model comparison p-value (Bonferroni corrected) |
| **CAPNS1** chr 19q13.12 | 0.990 | $5 \times 10^{-19}$ | **PPP1CC** chr 12q24.11 | 0.965 | $1 \times 10^{-36}$ |
| *COPS5* chr 8q13.1 | 0.935 | $4 \times 10^{-24}$ | RAB1A chr 2p14 | 0.845 | $2 \times 10^{-12}$ |
| *CSNK1A1* chr 5q32 | 1.000 | $1 \times 10^{-36}$ | **SHOC2** chr 10q25.2 | 0.995 | $1 \times 10^{-28}$ |
| *CUL5* chr 11q22.3 | 0.995 | $4 \times 10^{-27}$ | **TMEM147** chr 19q13.12 | 0.985 | $1 \times 10^{-24}$ |
| **DCTN6** chr 8p12 | 0.940 | $2 \times 10^{-27}$ | TREM2 chr 6p21.1 | 0.990 | 0.14 |
| *DSTN* chr 20p12.1 | 0.935 | $2 \times 10^{-33}$ | UBL3 chr 13q12.3 | 1.000 | $4 \times 10^{-33}$ |
| *EFNB3* chr 17p13.1 | 0.820 | $2 \times 10^{-4}$ | **ZNF207** chr 17q11.2 | 0.995 | $3 \times 10^{-27}$ |

step over this approach, we conducted set-based association testing for rare, deleterious SNVs with case/control status in ADNI. Gene sets were defined from the results of stability selection: each selected gene was grouped with its first neighbours in the hippocampus network. S2 Table reports association p-values with case-control status in ADNI for (A) burden, (B) variance-component, and (C) omnibus test (SKAT-O) for rare variants in the gene sets defined around selected genes. The gene grouping around *PFAS* was not associated with case-control status.

## Model comparison

The selection of genes whose smoothed scores correlated with case-control status was performed without considering the relative contribution of other established confounders such as sex and age. Therefore we sought to assess whether or not the smoothed scores carry any additional predictive power on top of the variation captured by such confounders, and to what extent. In ADNI, an extended logistic regression model including the smoothed score for *PFAS* significantly improved goodness-of-fit to case-control status over a baseline model including established AD predictors (sex, age, APOE ε4 count, education, population structure; chi-squared test p = $2x10^{-4}$). Additionally, the extended model showed a higher pseudo-$R^2$ statistic (p-$R^2$ = 0.36) than the baseline model (p-$R^2$ = 0.33), approximately equivalent to an additional 3% variance explained in the outcome.

In ADSP, we found that *CSNK1A1* was a second neighbour of *PFAS*, had a selection probability of 1 and provided the best improvement in goodness-of-fit to case-control status over the baseline model including sex, age and APOE ε4 (chi-squared test corrected p = $1x10^{-36}$; Table 1).

## Survival analysis

We postulated that the smoothed scores possess unique properties in that they condense in a biologically meaningful way information on how SNVs in a neighbourhood interact, acquiring an increased sensitivity to disease phenotypes. We therefore set out to further characterise the properties of such scores by assessing their relationship not only to simple case-control status but also to risk of clinical disease progression to AD.

The "raw" (binary) SNV status and smoothed score for *PFAS* in ADNI were tested for association with risk of conversion to AD using Cox proportional hazards model. The binary SNV status of *PFAS* was not seen to influence the probability of conversion to AD (p = 0.65; S9A Fig). The inclusion of covariates (see Methods) confirmed this result ($p_{PFAS}$ = 0.82; S9B Fig). Conversely, the smoothed score of *PFAS* showed a statistically significant protective effect against conversion to AD ($p_{PFAS}$ < 0.001; Fig 4) on top of the effect of other established confounders. The score was still significantly associated when removing healthy controls from the survival analysis to partially avoid circularity issues (p = 0.008; S9C Fig).

## Gene set enrichment analysis

Next, we examined the community of genes surrounding *PFAS* through gene set enrichments analysis, to see if *PFAS* can be regarded as hub of a module enriched in genes related to AD. Significant overlap was found for the 1,449 genes of interest with 1,815 gene sets from Gene Ontology (GO), Chemical and Genetic Perturbations (CGP), KEGG and REACTOME, at FDR of 5%. Enrichment p-values for eight gene sets related to AD are reported in S4 Table. A high-level visual representation of 928 significantly overrepresented GO terms in their semantic space can be found in S10 Fig. The KEGG AD pathway and the four sets curated by Blalock et al. [27] comprising dysregulated genes, were all significant at FDR 5%. Moreover,

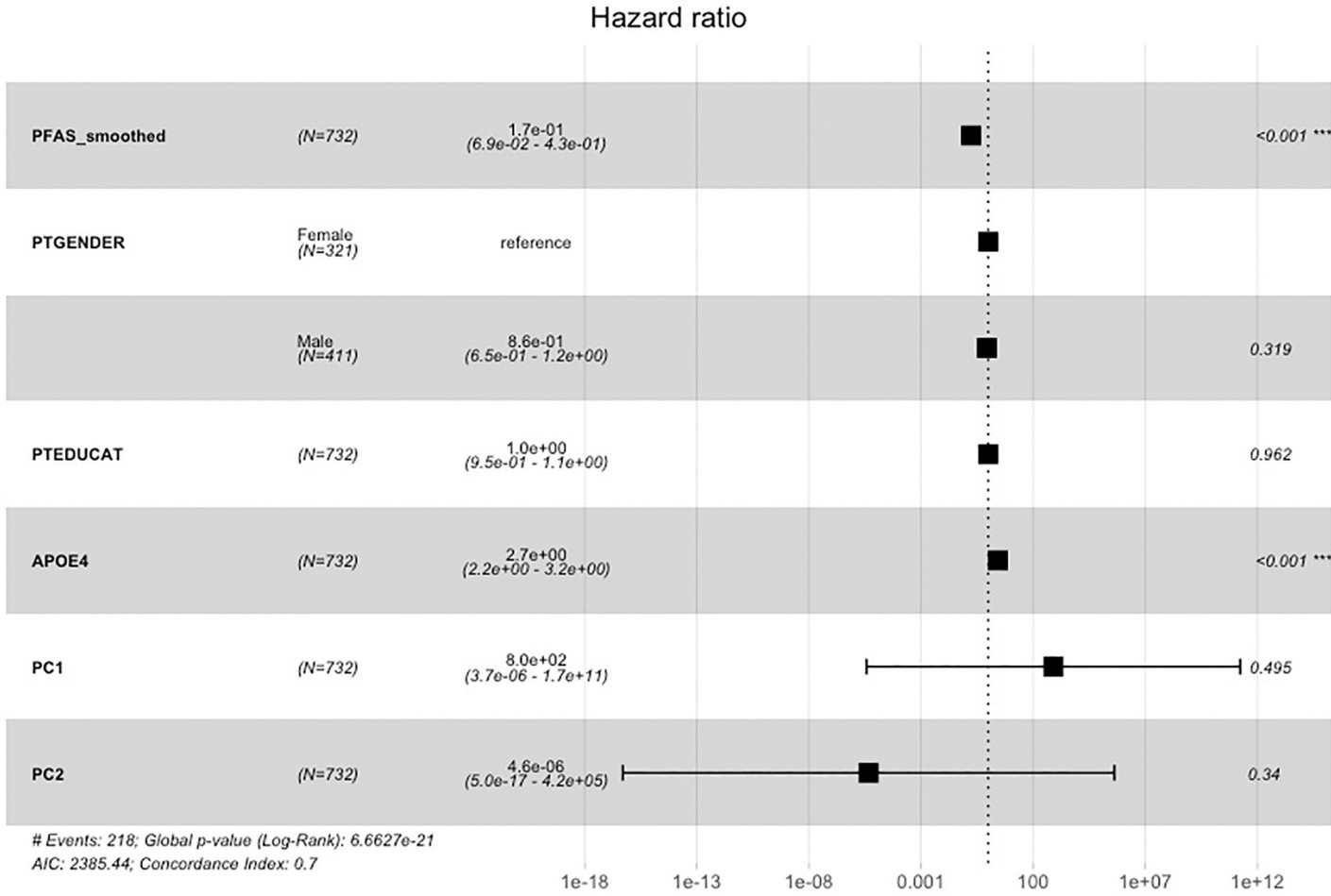

**Fig 4. Survival analysis with the smoothed score for the gene resulting from stability selection on the ADNI dataset.** The forest plot depicts hazard ratios from Cox proportional hazards model with confidence intervals and statistical significance. The score resulting from network propagation for *PFAS* was seen to be significantly associated with lower risk of conversion to AD.

randomisation control showed that the overlaps observed with the four Blalock sets could not be achieved by chance, as none of the randomly drawn gene sets returned uncorrected enrichment p-values lower than the ones observed, and only 14 out of 1,000 random sets achieved more significant overlap with the set of genes downregulated in AD curated by Wu et al. [28] (S11 Fig).

## Differential gene expression analysis

Motivated by the observed overlaps with sets of genes dysregulated in AD, we lastly sought to investigate whether the percolation of rare variants' effects through the network impacts the transcriptomic profiles of the 30 genes identified in ADNI and ADSP, and furthermore, whether this impact exhibits tissue specificity. To test this hypothesis, we leveraged two independent, publicly available RNA-seq datasets: temporal cortex samples from the Mayo Clinic Brain Bank, and parahippocampal gyrus samples from the Mount Sinai Brain Bank.

In the Mayo RNA-seq dataset for temporal cortex, none of the investigated genes showed significant dysregulation between controls and AD cases after correction for cell type gene markers. Full results and visualisations of the differential expression analysis in the Mount

Sinai dataset for parahippocampal gyrus are reported in S5 Table and S12 Fig. In this dataset, we observed a trend towards downregulation for *PFAS* and *SHOC2* in controls vs definite AD ($p_{PFAS}$ = 0.04; $p_{SHOC2}$ = 0.001); however these did not survive multiple comparisons correction. Sixteen genes among the 30 analysed were found to be differentially expressed in at least one pairwise comparison of disease severity. A randomisation control demonstrated that this number is significantly higher than it would be expected by chance (p < 0.001; S13 Fig). In particular, significant dysregulation in controls vs definite AD was observed for *ARL1*, *ATXN10*, *CAMK2B*, *CAPNS1*, *EFNB3*, *GNB1*, *KCNMA1*, *KLC1*, *MAPRE1*, *MAPRE3*, *MOB4*, and *TREM2*. The tissue specificity of the differential expression results (i.e., dysregulation seen in parahippocampal gyrus but not in the temporal cortex as a whole) can be interpreted as a reflection of the use of a tissue-specific gene interaction network (hippocampus) and of the high spatial variability in brain transcriptomic profiles.

## Discussion

In this work we presented NETPAGE, a computational tool for gene-based association testing of rare variants that integrates prior knowledge about tissue-specific gene interaction networks. The aim was to boost the information content assigned to each gene, by enriching it with the knowledge of the pathways through which rare variation percolates. NETPAGE leverages the well-known strength of network propagation, combining it with multivariate sparse regression to identify genes robustly associated with a disease phenotype by analysing the genome-wide landscape as a whole and without the need for stringent statistical corrections.

NETPAGE allowed us to reveal correlations between the propagation of rare variant effects at the gene level and disease outcomes (both diagnosis and risk of conversion), as well as tissue-specific downstream effects of such propagation at the transcriptomic level. We tested the proposed method on a small dataset (ADNI), as well as on a medium-sized sample (ADSP). This enabled us to put the spotlight on a set of genes mutually connected in the hippocampus network and belonging to the same neighbourhood of diameter two. We were also able to demonstrate robustness to the choice of network by replicating the association with *PFAS* with the STRING PPI network. But one of the core features of NETPAGE is its flexibility and applicability to a wide range of traits and diseases whose genetic architecture follows the trend described by Manolio *et al.* [2].

We investigated the behaviour of the network propagation step in NETPAGE through simulated data for a range of SNV frequencies, and optimised some of the key parameters of the diffusion process. As expected, when the neighbouring genes' SNV frequencies do not differ between cases and controls, no difference is detectable in the smoothed score of the hub gene, whereas different SNV frequencies in the neighbouring genes always flow into the hub to determine a smoothed score significantly different between cases and controls, even when the hub gene itself is not mutated, under a range of parameters and mutation scenarios. In agreement with Hofree *et al* [19], we also found the diffusion length α to have a minor, if not negligible, effect on the hub gene's smoothed score over a sizable range and in all mutation scenarios considered. However, in contrast to Hofree et al., we found the final quantile normalisation step to be detrimental to the process and therefore excluded it from our applications to real data.

Since we regard NETPAGE as a gene-based association testing tool, we compared its performance against SKAT-O, a state-of-the-art method for gene- and set-based association testing of rare variants, in both datasets. In contrast to Bis *et al.* [11], SKAT-O on ADSP did not identify any gene-wide significant gene (S8 Fig). Unexpectedly, not even *TREM2* showed a gene-wide significant association in our SKAT-O analysis of ADSP, despite being nominally

significant ($p = 9\text{x}10^{-3}$). We are inclined to interpret this result as a consequence of different annotation pipelines and our selection criteria being somewhat stricter (despite applying the same CADD threshold), leading to a drastic reduction in the number of SNVs to be grouped and tested (270,165 here vs 918,053 variants in Bis *et al.* [11]), and likely to result in much lower burden test statistics. Additional negative controls in ADNI compared NETPAGE to mass-univariate testing of smoothed scores, stability selection on the "unsmoothed" SNV burden, and evaluated the impact of different networks (random, non-brain, and other brain components) on stability selection. None of these techniques reported any association at the gene-wide, Bonferroni-corrected significance threshold or at the selection probability threshold of 80% (S3 and S7 Figs and S2 Table), and the gene selection was also tissue-specific in brain (S4 Fig), clearly demonstrating the added value provided by our approach. On the subject of tissue-specificity, it is important to note that there is no "correct" or "incorrect" tissue network in an absolute sense, but the relevance of the network needs to be evaluated in reference to the disease under investigation. By using networks for non-brain tissues and other brain regions, we intended to show that results change by tissue and that, when a tissue unrelated to Alzheimer's disease is investigated (such as the medulla oblongata in S4 Fig), the selected gene(s) do not show biological relevance to the disease. In the case of *ZBBX*, this gene does not map to any known pathways and we could not find evidence in the scientific literature of links with Alzheimer's disease.

There are many network propagation-based methods proposed over the last years, such as CATAPULT [29], HotNet2 [30], or NBS [19]. However, benchmarking NETPAGE against these methods is not a straightforward task, since these methods were designed for purposes fundamentally different from association testing, namely gene prioritisation, module detection, or patient stratification, respectively. There are also several recent studies where network propagation has been leveraged to boost power of genetic association testing with remarkable improvements [31–33]; however, these applications are substantially different from NETPAGE, in that the focus there was on common variation (GWAS datasets) while here we set ourselves the more challenging task of investigating rare variants in NGS experiments.

We also investigated the effect of different selection criteria to include SNVs in the rare variant burdens to be propagated through the network. When we focused on stop-gain, stop-loss and frameshift insertions to derive our gene burdens for network propagation, we did not detect any association signal. This type of rare variants has generally a higher functional impact than missense variants, and is therefore usually subject to stronger selective pressure due to a dramatic decrease in fitness. This translates into even sparser SNV profiles, whereby the "signal enhancement" capability of network propagation is still not sufficient to reveal separation between cases and controls. We conclude that the inclusion criteria for SNVs–which are not to be regarded as a parameter of the algorithm underlying NETPAGE but rather as a user choice–need to reflect careful consideration of the disease area under study as well as the experimental conditions (sample size, sequencing technology and the like).

The genes identified by stability selection exhibit connections to a number of intermediate phenotypes and biochemical processes relevant to AD. For instance, there are multiple, distinct lines of evidence linking *PFAS* (phosphoribosylformylglycinamidine synthase) to Alzheimer's Disease. Common missense variation in *PFAS* has been linked to low-density lipoprotein (LDL) cholesterol measurement through GWAS [34,35], and elevated LDL cholesterol is reportedly a risk factor for AD, through an increased production of beta-amyloid protein [36]. Additionally, Gene Ontology annotations for *PFAS* include the term 'glutamine metabolic process' (GO:0006541). The metabolism of glutamine and the glutamine-glutamate cycle takes place in the brain at the astrocyte level [37], and disruptions in this cycle leading to glutamate excess in the synaptic cleft trigger neuronal excitotoxicity, which has been

implicated in AD [38]. But most importantly, PFAS is involved in the purine metabolism pathway (see for instance the KEGG pathway hsa00230), whose purpose in humans is to maintain an optimal level of the nucleotides in the tissues. Several studies have implicated perturbations in purine metabolism in the mechanisms of neurodegenerative disorders, including AD. For instance, Xiang et al. [39] found evidence for a role of the purine metabolic pathway through an integrative approach bringing together GWAS, KEGG pathways and gene expression; proteomic profiling in Kaddurah-Daouk et al. [40] showed that the purine pathway is directly related to CSF total tau; Ansoleaga et al. [41] showed stage- and region-dependent deregulation of purine metabolism in AD by analysing mRNA expression levels of purine metabolism genes in entorhinal cortex samples (the entorhinal cortex surrounds the hippocampus and is one of the earliest regions to be affected by tau accumulation in AD). Lastly, González-Domínguez et al. [42] showed metabolic alterations compatible with deregulation of the purine pathway mainly localised in the hippocampus of the APP/PS1 mouse model of AD. Hence it seems plausible that rare deleterious variation in PFAS or its functional neighbourhood should increase susceptibility to AD through its impact on cholesterol and glutamate levels as well as the synthesis and metabolism of purines.

The association of rare, loss-of-function SNVs in *TREM2* with AD is already well-established [43]; a discovery that historically paved the way to crucial findings about the role of neuroinflammation, microglia and innate immunity in AD [44–46]. Additionally, interactions for other NETPAGE-selected genes such as *APPBP2* and *CAPNS1* have been reported with the amyloid-precursor protein *APP* [47,48]. Furthermore, the local gene-interaction neighborhood of *CAMK2B*, which was identified by NETPAGE, shows a series of compelling dementia risk genes (Fig 3C): key players in familial AD (*APP*, *PSEN2*), genes in which rare variation is linked to sporadic AD (*PLD3*, *ABCA7*, *SORL1*; [49–51]), GWAS genes (*APOE*, *CLU*, *BIN1*, *PICALM*), genes identified through multivariate imaging genetics studies (*TRIB3*; [52]) and genes causal for other neurodegenerative diseases (*MAPT*, *GRN*, *HTT*, *OTUD4* [53]). The selected hub genes therefore recapitulate the biology of some of the most important pathological processes and risk factors not only for AD, but for a broadly-defined neurodegenerative phenotype.

We demonstrated the successful application of NETPAGE to two independent datasets, one of small sample size (ADNI) and the other of moderate sample size (ADSP). In conducting these experiments, we did not seek to achieve replication of the selected genes; this task is an intrinsically challenging one, given the rarity of the SNVs considered, the difference in coverage (an additional 3,000 genes were tested in ADSP that were not present in ADNI), and the small sample considered in ADNI. Our main focus was to demonstrate the validity of the proposed method in application to two very different, real-world experimental scenarios. However, it is indeed remarkable that short-range functional connections link a subset of the genes selected in the two datasets in a statistically significant fashion (S6 Fig), and we believe this to be a powerful proof of the working hypothesis underlying NETPAGE.

The role of *PFAS* as hub of a molecular pathway disrupted in AD is further strengthened by the finding of extensive and significant overlaps between the subnetwork centred on *PFAS* and the KEGG AD pathway, as well as with curated sets of genes affected by transcriptional dysregulation at distinct stages of the disease (S5 Table and S11 Fig). Additionally, the neighbourhood of *PFAS* appears to be enriched in genes related to a number of key ontology terms (S10 Fig), among which: biological processes such as cell ageing, microtubule-based processes, and mRNA metabolism; and cellular components such as mitochondria, myelin sheath, axon cytoplasm, and ribonucleoprotein complexes [54]. These overlaps provide additional support to the hypothesis that rare deleterious variation percolates through the network structure to significantly alter the protein landscape of the cellular environment in a pathological way. However, we also showed that these alterations are tissue-specific (and most likely even cell type-specific),

since we reported differential expression for some of the core genes identified in the parahippocampal gyrus from the Mount Sinai dataset but not in samples from the broader temporal cortex in the Mayo RNA-seq dataset, at different disease stages (S5 Table). More importantly, we observed transcriptional alterations in relation to AD diagnosis in more than half of the putative core genes investigated, and demonstrated that this effect could not have been observed by chance (S13 Fig). This finding can be interpreted as pointing towards a co-expression module altered by disease. This further supports our idea that integrating information about variation and interactions is a powerful approach to gain a comprehensive, systems-level view of disease-related molecular mechanisms, as opposed to the investigation of single variants or genes of interest. However, as we did not detect differential expression for all the genes identified, we speculate that the link between these putative core genes and AD might reside in molecular mechanisms other than RNA or protein abundance, such as splicing, regulation or post-translational modifications. Some evidence in this direction is provided by a significant enrichment in genes related to post-transcriptional regulation of gene expression (S10 Fig, top panel).

We proposed NETPAGE, a methodology that enables the exploration of biological pathways through which structural variation affects tissue function and results in complex disease phenotypes. The rationale followed by NETPAGE resembles aspects of the recently proposed, although debated, omnigenic model [14,55]. There remain, however, some limitations. First, an inherent difficulty is that the ADNI dataset is a low-coverage WGS, while ADSP is WES, which can be biased due to the required exome enrichment step. Therefore, on one hand ADNI might not provide a very clean signal, while on the other hand ADSP might not enable a strict validation due to the different sequencing methods utilised. Second, we restricted this initial study design to include only rare exonic SNVs whose deleteriousness was assessed through the CADD score. Extensions of this work may address the impact on gene discovery of choices related to the study design, such as the CADD threshold for deleterious SNVs inclusion or the inclusion of intronic and regulatory variants. Motivated by the observation that the selected genes are all hubs of the hippocampus network (S3 Table), we envisage that future method developments may also include an alternative test, where the score from network propagation is compared to an expected score based solely on gene degree. Finally, the use of an established sparse regression framework makes NETPAGE a highly flexible method, therefore future work may explore the relationship between smoothed rare variant profiles and AD-related continuous traits, such as imaging or fluid biomarkers, beyond the simple binary definition of diagnosis.

In summary, we demonstrated a novel application of network propagation to the study of rare variant effects on complex traits in two sporadic, late-onset AD cohorts. NETPAGE allowed us to identify a set of genes as tightly interconnected network hubs where the downstream influence of rare variants accumulates and acquires predictive power for diagnosis, as well as to provide multiple lines of evidence for the biological meaning of the smoothed scores and the tissue-specific involvement of some core genes at the transcriptional level. We emphasise the flexibility of the presented methodology, that enables enhanced association testing for binary and continuous traits, as well as small and large sample sizes, as NGS is becoming an increasingly affordable alternative to SNP genotyping and adopted in many cohort and biobank studies. We believe NETPAGE is a promising approach for determining novel genetic influences on complex traits and for providing mechanistic insights into disease biology.

## Materials and methods

### Ethics statement

The use of off-the-shelf de-identified human subject information from the Alzheimer's Disease Sequencing Project (ADSP) was conducted at Stanford University and was exempt from

Institutional Review Boards approval. The analysis of off-the-shelf de-identified human subject data from the Alzheimer's Disease Neuroimaging Initiative (ADNI) was conducted at UCL and was exempt from ethics approval. The analysis of off-the-shelf de-identified human subject data from the Mayo Clinic and Mount Sinai Brain Bank was conducted at UCL and was exempt from ethics approval, as all data is publicly available for download from Synapse.

## ADNI WGS data preprocessing

Data used in the preparation of this article were obtained from the Alzheimer's Disease Neuro-imaging Initiative (ADNI) database (adni.loni.usc.edu) and the Alzheimer's Disease Sequencing Project (ADSP) [7]. The ADNI was launched in 2003 as a public-private partnership, led by Principal Investigator Michael W. Weiner, MD. The primary goal of ADNI has been to test whether serial magnetic resonance imaging (MRI), positron emission tomography (PET), other biological markers, and clinical and neuropsychological assessment can be combined to measure the progression of mild cognitive impairment (MCI) to early AD. For up-to-date information and a complete list of investigators, see www.adni-info.org and S2 Text. The ADNI WGS dataset was used for application to a small sample size. SNVs and small insertions-deletions (indel) data was available for 808 ADNI subjects. WGS data in Variant Call Format (VCF) was downloaded from the ADNI database. AD case or control status was available for 476 out of 808 individuals (for the remaining participants the latest diagnosis available was of MCI). Demographics and clinical outcomes for this sample are presented in Table 2.

## ADSP WES data preprocessing

The ADSP WES dataset (N = 10,913) was then used for application to a moderate sample size. We identified 24 samples who were also sequenced as part of ADNI, who were then removed from the ADSP WES dataset, yielding a sample size of 10,889 (S1 Text). Demographics and clinical outcomes for this final sample are presented in Table 2.

## Ancestry and population structure

Ancestry and population structure on the ADNI dataset were previously analysed from GWAS data, using SNPweights version 2.1 [56] and a two-step procedure described in [57]. We leveraged the results of this analysis to retain only study participants showing a probability of being of Caucasian ancestry greater than 80% (N = 439 AD cases and controls). For these subjects we also obtained principal components of population structure to be used in subsequent analyses.

In ADSP, due to the absence of GWAS data, individuals were filtered based on self-reported ethnicity and racial background, to include only Caucasian participants (self-reported white race and not hispanic or latino ethnicity; N = 10,413). Lastly, case-control status was available for 10,186 of these Caucasian individuals.

**Table 2. Demographics and clinical outcomes for the ADNI WGS case/control sample and the ADSP WES case/control sample.**

| | ADNI | | | ADSP | | |
|---|---|---|---|---|---|---|
| | HC | AD | Total | HC | AD | Total |
| **Sample size** | 246 | 230 | 476 | 4,796 | 5,852 | 10,648 |
| **Women** | 134 | 92 | 226 | 2,819 | 3,371 | 6,190 |
| **White, not hispanic or latino** | 222 | 217 | 439 | 4,601 | 5,585 | 10,186 |
| **Age at latest visit (mean ± sd)** | 78.3 ± 7.27 | 79.1 ± 7.58 | 78.69 ± 7.42 | 86.1 ± 4.53 | 75.4 ± 8.45 | 80.19 ± 8.76 |
| **APOE ε4 (0/1/2)** | 181/64/4 | 80/117/33 | 261/181/37 | 4,080/701/15 | 3,379/2,314/159 | 7,459/3,015/174 |

## Variant filtering and mapping

We used ANNOVAR, 01/06/2017 release [58] to annotate the SNVs, retaining only exonic, non-synonymous SNVs, with MAF <1% in non-Finnish-Europeans from the Exome Aggregation Consortium [59]. Variants were further filtered based on deleteriousness, by retaining SNVs ranked among the top 1% according the Combined Annotation-Dependent Depletion (CADD) method (CADD phred score>20) [60]. Exonic SNVs were mapped to genes according to annotations based on the RefSeq database [61]. These filtering and mapping procedures were applied to the ADNI and ADSP datasets separately. This resulted in 48,834 rare, non-synonymous SNVs mapped to 13,591 genes in ADNI, and 270,165 rare, non-synonymous SNVs mapped to 16,630 genes in ADSP.

To assess the impact of the filtering criteria outlined above we constructed a second set of SNVs, by retaining exonic variants with MAF < 1% predicted to have a stop-gain, stop-loss or frameshift consequence. We also applied the CADD phred > 20 filter to stop-gain and stop-loss variants, and performed mapping to RefSeq genes.

## Tissue-specific gene interaction networks

As a substrate for network propagation, we leveraged tissue-specific weighted gene interaction networks from Greene et al. [15]. In these networks, each node represents a gene, each edge a functional relationship, and an edge between two genes is probabilistically weighted based on experimental evidence connecting both genes. We focused on the interaction network for the human hippocampus, being the key brain structure related to the loss of episodic memory in AD [62]. Additional negative controls were performed using: 30 replicates of a degree-preserving randomised version of the same hippocampus network, obtained by adapting a publicly available python implementation by C. Lasher (https://gist.github.com/gotgenes/2770023#file-edgeswap-py); and a gene network not related to brain tissue, specifically the umbilical cord network. We also performed network propagation in different brain structures or components (neuron, medulla oblongata, diencephalon) to investigate the specificity of association results to AD-related brain regions. Lastly, we investigated the robustness of the method with respect to the network structure and derivation methods by using the human, non-tissue-specific protein-protein interaction network (PPI) available through the STRING database [21] and the recently published human 'all-by-all' reference interactome map of human binary protein interactions, or 'HuRI' [63].

## Propagation of rare variant signals through gene interaction networks

To model the propagation of the effects of rare variants through a gene interaction network, we adopted the *network propagation* approach first introduced by [64] for semi-supervised learning. In our case, a gene carrying a deleterious variant is used as a seed in an iterative procedure that propagates its effect according to the network structure. Hence the effect propagation effectively reproduces a graph-constrained diffusion process whereby information from the seed gene flows not only to its first neighbours but to all genes in a connected component, and the amount of information flowing into a gene is determined by the strength of its connections to the source gene.

Practically, a single individual's whole genome data is represented as a gene-based vector of length M, where for each gene the rare variant burden can be encoded as either the count of rare deleterious variants (rare-variant burden) or a binary variable indicating the presence or absence of any such variant (rare-variant status). Stacking these vectors of all S subjects yields the sparse matrix $G^0 \in \mathbb{N}^{S \times M}$. A network can be either in the form of a weighted graph,

represented by a square, symmetric, similarity matrix $N \in \mathbb{R}^{M \times M}$ with $0 \leq N_{ij} \leq 1$, or in the form of an unweighted graph, represented by a square, symmetric adjacency matrix $N \in \mathbb{N}^{M \times M}$ with $N_{ij} \in \{0; 1\}$. The gene burden $G^0$ is propagated through the network simultaneously for all subjects according to the following iterative procedure:

$$G^{t+1} = \alpha G^t \cdot (D \cdot N) + (1 - \alpha) G^0 \tag{1}$$

where $G^{t+1}$ is the smoothed rare variant profile at iteration $t+1$; $\alpha \in [0,1]$ is a tuning parameter governing the distance a signal is allowed to diffuse through the network, and $D \in \mathbb{R}^{M \times M}$ is a diagonal matrix with the inverse of the node strengths of $N$ along the diagonal. Eq (1) is iteratively evaluated until convergence (i.e., until the $L^2$-norm of $G^{t+1} - G^t$ is smaller than a pre-specified threshold).

Our implementation of network propagation in NETPAGE allows the user to specify: whether the network used is a gene- or a protein-interaction network, and the gene/protein naming convention; the type of encoding used in the input file (i.e., if rare-variant burden or rare-variant status is used); the convergence threshold on $\|G^{t+1} - G^t\|_2$ (default $10^{-6}$); the diffusion length $\alpha$ (default 0.5); this was optimized through simulation; whether the full weighted graph is to be used to guide the diffusion process, or if a graph adjacency matrix is to be generated from the original graph by retaining only the top P% edges (we refer to this procedure as network binarisation); the percentage P of top edges to be retained, in case network binarisation is to be performed (default 1%); this was optimized through simulation; whether the gene interaction network features self-loops (default False); whether the rows of the smoothed rare variant profile are to be quantile-normalised after convergence (default False). Quantile normalisation is the last step performed after network propagation by NBS [19], to ensure that the smoothed rare variant profile for each patient follows the same distribution.

Network propagation is sensitive to the direction of rare variant's effect (i.e., protective vs. deleterious), therefore the user can choose if and how to deal with the direction of effects of the genes' rare variant burdens with respect to a given binary phenotype. Briefly, the bioinformatics assessment of deleteriousness (e.g., CADD score) is unrelated to any disease phenotype, hence the rare variant status of a given gene can be equally risk-increasing or protective with respect to a specific binary phenotype, resulting in blended effects in signal-receiving genes. Therefore, we require a mechanism to numerically distinguish the "signal" flowing into a hub from a protective gene from the "signal" coming from a risk gene. In light of this, we are providing the user with additional flexibility to either: set to 0 the rare variant status/burden of protective genes; set to 0 the rare variant status/burden of risk genes; set the rare variant status to -1 for protective genes and to +1 for risk genes; none of the above. Risk and protective genes are determined by the direction of effect of their rare variant status on the case-control status (odds ratio from a Fisher's test on the 2x2 contingency table).

A Python 2.7 module implementing network propagation for this study is available at https://github.com/maffleur/NETPAGE.git.

## NETPAGE: application to AD sequencing data

We performed network propagation on the full (i.e., without filtering on ancestry or diagnosis) ADNI and ADSP datasets separately. Each subject-level rare variant burden (row of the $G^0$ matrix) was encoded as a binary vector of length M (0 = gene not carrying any of the selected SNVs, 1 = gene carrying at least one of the selected SNVs). The $G^0$ matrix in ADNI described the rare-variant status of 13,310 genes for 808 study participants. The $G^0$ matrix in ADSP described the rare-variant status of 16,268 genes for 10,889 study participants. In both applications, we used the default values for the parameters $\alpha$ and P, as optimised through simulations, and set the rare-variant status to -1 for protective genes and to +1 for risk genes.

## Stability selection

In order to identify genes robustly associated with disease, smoothed rare variant profiles resulting from network propagation were related to clinical diagnosis via sparse logistic regression. We applied LASSO regression [22] as implemented in the R package *glmnet* [65]. Briefly, *glmnet* finds the coefficients' vector β that solves the following regularised regression problem:

$$min_{\beta_0, \beta} \frac{1}{N} \sum_{i=1}^{N} l(y_i, \beta_0 + \beta^T x_i) + \lambda |\beta|_1 \tag{2}$$

over a range of values for λ. Here $l(y, y_{est})$ is the negative log-likelihood contribution for observation *i*. In Eq (2), $N$ is the number of observations (subjects), $\beta_0$ is the model intercept and $\beta$ the vector of regression coefficients. For subject *i*, $y_i$ is the value of the response variable, and $x_i$ a vector of predictors. The LASSO or L1-regularisation corresponds to minimising the L1-norm of the coefficients' vector $|\beta|_1$. As a consequence, most of the coefficients in $\beta$ shrink to zero, achieving efficient variable selection. The amount of regularisation is controlled by $\lambda$, also known as sparsity hyperparameter. Here, we performed stability selection [23], as implemented in the R package *stabs* [66], to automatically tune the sparsity hyperparameter while at the same time performing feature selection. In brief, stability selection combines variable selection with bootstrap resampling to estimate a probability value for each variable to be selected by the sparse regression. For a given regression task, we performed 100 bootstrap (split-half) resamplings and focused on smoothed gene scores with a selection probability higher than 80% over a range of values for the sparsity hyperparameter $\lambda$. Stability selection in ADNI was performed on smoothed scores for M = 13,310 genes and N = 439 Caucasian subjects (222 healthy controls [HC], 217 AD; date accessed April 24th, 2018). Stability selection in ADSP was performed on smoothed scores for 16,298 genes and 10,186 Caucasian subjects (4601 HC, 5585 AD).

## Gene-based rare variant association testing

We conducted gene-based association testing for rare, deleterious SNVs with clinical diagnosis in ADNI with a state-of-the-art method, in order to benchmark NETPAGE's performance. Specifically, we used SKAT-O [25], which combines burden and variance-component tests, implemented in the R package *SKAT* [10]. Diagnosis at the latest available time point was coded as a dichotomous trait (HC vs AD). Associations were tested for M = 13,591 genes controlling for age, sex, number of *APOE* ε4 alleles, years of education, and two principal components of CEU population substructure (see section *Ancestry and population structure*). Genome-wide significance was established at p < 0.05/13,591 = 3.6x10⁻⁶. Additionally, to demonstrate the advantage of our multivariate approach, a mass univariate test of association between the smoothed gene scores and case-control status was also conducted via logistic regression for M = 13,310 genes, controlling for the same confounders listed above. Genome-wide significance was established at p < 0.05/13,310 = 3.75x10⁻⁶. The sample for both these tests comprised 439 Caucasian cases and controls from ADNI.

We further conducted a gene-based association test with SKAT-O on the same 10,186 European subjects from ADSP used for stability selection; 270,165 non-synonymous SNVs were grouped into 16,630 genes. Associations were tested correcting for age, sex, number of APOE4 alleles, and sequencing centre. The aim of this test was to benchmark NETPAGE against a state-of-the-art method on a dataset that enables sufficient statistical power to detect associations, as it has been already reported [11].

### Set-based rare variant association testing

We conducted set-based association testing for rare, deleterious SNVs with case/control status in ADNI. Gene sets were defined from the results of stability selection: each selected gene was grouped with its first neighbours in the hippocampus network. Burden, variance-component, and omnibus (SKAT-O) tests were performed with this set definition.

### Model comparison

After stability selection, we focused on the smoothed score of the top selected gene. With it, we built two logistic regression models: a baseline model, including case-control status as response, and as predictors the same set of covariates used for SKAT; and an extended model, adding to the predictors in the baseline model the smoothed score of the selected gene. Sample size for these models was 439 Caucasian cases and controls from ADNI. We compared the goodness-of-fit of the two models through a chi-squared test. Additionally, we computed the pseudo-$R^2$ statistic (the analogue of the percentage of variance explained in linear regression models) for the baseline and the extended model. We used the McFadden method in the Pseudo-R2 function in the R package *DescTools* [67]. We repeated the same procedure for the ADSP-selected genes, one at a time. P-values from the chi-squared tests were corrected for multiple comparisons using the Bonferroni procedure (Table 1). The predictors for the baseline model now included only sex, age and number of *APOE* ε4 alleles. Years of education was not recorded among the ADSP phenotypes. It was also not possible to compute CEU population substructure, due to lack of GWAS data. Sample size for these models was 10,186 Caucasian cases and controls.

### Survival analysis

We conducted survival analysis on the selected genes in ADNI with the R packages *survival* [68] and *survminer* [69]. The event considered was either conversion from HC to AD or conversion from MCI to AD. The time variable was age at dementia onset if conversion occurred; or age at latest diagnosis if conversion did not occur (right-censored time-to-event). Sample size was N = 732 ADNI Caucasian subjects with WGS data (>80% CEU ancestry). For the selected genes we considered both the "raw" (0/1) rare variant status, and the smoothed score derived with network propagation. For each gene, we first modelled and compared survival curves for SNV carriers vs non-carriers (log-rank test); we then refined the comparison of carriers and non-carriers by fitting Cox proportional hazards models, controlling for sex, number of *APOE* ε4 alleles, years of education, and two principal components of population. Lastly, we fitted the same Cox models replacing the binary rare-variant status for the gene with its smoothed score, retaining the same set of covariates. Because healthy controls were already used for stability selection, in order to mitigate the risk of a circular analysis we repeated the analysis excluding stable healthy controls ($N_{HC}$ = 222, final sample size N = 510), therefore focusing only on stable MCI who did not convert to AD as the event-free population.

### Gene-set enrichment analysis

Gene set enrichment analysis was performed on the gene selected in the ADNI dataset, together with its first and second neighbours in the thresholded and binarised hippocampus gene network. This yielded a set of 1,449 genes to be tested. As background set, we used the set of M = 13,310 genes resulting from network propagation on the ADNI dataset. We tested for enrichment using the Gene Ontology (GO) and chemical and genetic perturbation (CGP) gene sets, as well as the KEGG and REACTOME pathway collections available in the

Molecular Signatures Database v6.1 [70]. Overrepresentation of the selected genes in these curated sets was tested with the Fisher's exact test. Enrichment p-values were adjusted for multiple comparisons using the Benjamini-Hochberg procedure at a false discovery rate of 5%. Significant GO terms were aggregated in hierarchies and visualised through ReViGO (http:// revigo.irb.hr/revigo.jsp) [71]. For seven CGP gene sets relevant to Alzheimer's disease, we further applied a randomisation procedure to ensure that the observed enrichments could not be achieved by chance. We formed 1,000 set replicates by randomly drawing 1,449 genes from the background; every set replicate was tested for overrepresentation in the seven CGP sets, then the p-values from the original set were compared to the distributions of p-values from the randomised sets.

### Differential gene expression analysis

Motivated by the observed overlaps with sets of genes dysregulated in AD, we lastly sought to investigate whether differential expression between cases and controls occurs for genes selected in ADNI and ADSP. In order to investigate also the tissue-specificity of this effect, we leveraged two independent RNA-seq datasets: the Mayo Clinic Brain Bank (MCBB) [72] and the Mount Sinai Brain Bank (MSBB) [73]. Within these datasets, we specifically focused on post-mortem expression levels in the temporal cortex and parahippocampal gyrus (Brodmann area 36), respectively. This choice aimed at maximising proximity and consistency with the hippocampal gene interaction network used in the discovery phase, being aware of the high degree of spatial variability in transcriptomics patterns across the brain. All data was accessed through the AMP-AD Knowledge Portal (www.synapse.org). For the Mayo dataset, we leveraged publicly available results of differential expression analysis, with and without correction for neuronal marker gene levels (Synapse ID syn6090802). RNA sequencing and processing at the MSBB was described in detail elsewhere [73]. Each sample was assigned a neuropathology category according to the Consortium to Establish a Registry for Alzheimer's Disease (CERAD) protocol (1 = normal, 2 = definite AD, 3 = probable AD, 4 = possible AD) [74]. We formed a gene set of interest comprising genes selected in ADNI and ADSP; we then used the Dunn's test for stochastic dominance [75] to perform pairwise nonparametric testing for differences in normalised gene expression levels between CERAD neuropathology categories in this gene set of interest. P-values were corrected for gene-wise, pairwise comparisons using the Benjamini-Hochberg procedure at a false discovery rate of 5%, and the Bonferroni method for the number of genes tested. Lastly, we performed a randomisation control to quantify the probability of seeing N genes with a significant differential expression among M genes randomly drawn from the MSBB dataset. We formed 1000 sets of M randomly drawn genes, conducted the Dunn's test for each gene and set, and counted how many genes in each random set showed significantly different expression in at least one pairwise comparison. We then plotted the distribution of the number of randomly dysregulated genes and compared it to the actual number of dysregulated genes found in the original set of interest.

### Supporting information

**S1 Text. Supplementary methods.**
(DOCX)

**S2 Text. Consortia acknowledgments.**
(DOCX)

**S1 Table. Selection probabilities (from the application of NETPAGE to ADSP) for the 21 genes reported in the recent GWAS by (Kunkle _et al._, 2019) [24].** If a gene did not have any

SNVs mapped to it, and therefore was not included in network propagation, this resulted in missing selection probabilities (NA values).
(DOCX)

**S2 Table. Results of set-based SKAT test for association of rare, exonic, deleterious variants with case-control status in ADNI.** The set of interest was formed by including the first interaction neighbours of PFAS. Additionally, the first two rows in the table report gene-based p-values for PFAS from S7 Fig.
(DOCX)

**S3 Table. Degree of the 30 genes selected in ADNI and ADSP in the hippocampus functional network.**
(DOCX)

**S4 Table. Results of gene set enrichment analysis for eight AD-related gene sets.** The gene set tested for overrepresentation included PFAS and its first and second interaction neighbours in the thresholded and binarised hippocampus functional network. Hits = number of observed genes overlapping with the curated gene set of interest; expected = number of genes expected to overlap with the curated gene set of interest by chance; OR = odds ratio from the Fisher's test; P = p-value of the Fisher's test; $P_{corr}$ = p-value corrected with the Benjamini-Hochberg procedure. P-values reported in bold are significant at FDR 5%.
(DOCX)

**S5 Table. Differential expression analysis uncorrected, two-tailed p-values on the MSBB RNA-seq dataset for the 30 genes selected from ADNI and ADSP.** The + or–sign in brackets next to the p-value represents the direction of effect detected. Pairwise comparisons were performed for normalised expression levels against CERAD diagnosis using the nonparametric Dunn's test. Values in boldface are the ones that remained significant after a two-fold multiple testing correction (Benjamini-Hochberg for the multiple pairwise comparisons for a given gene, Bonferroni for the number of genes tested). In total, 16 genes were seen to be significantly dysregulated in at least one comparison.
(DOCX)

**S1 Fig. A selection of simulation results. The x and y axis represent the mutation frequencies in controls and cases respectively**. The faceting allows to visualise the effect of other parameters (diffusion length, percentage of edges retained, quantile normalisation). The colour-coding indicates the statistical significance of the difference in the hub gene smoothed score between controls and cases, in units of -log10 p-value. We investigated three mutation scenarios: scenario 1, only first neighbours mutated; scenario 2, only second neighbours mutated; scenario 3, both first and second neighbours mutated; the target gene is always unmutated. Cells marked with a black cross indicate parameter combinations where the smooth score of the hub gene was not significantly different between cases and controls. (A) Effect of quantile normalisation after network propagation with top 1% edges retained (top row, scenario 1 on the left, scenario 3 on the right), and with top 25% edges retained (bottom row, mutation scenarios as in top row). (B) Joint effect of percentage of edges retained and diffusion length for the three mutation scenarios considered.
(DOCX)

**S2 Fig. Stability selection results in ADNI for a selection probability cutoff of 0.80, after propagating the burden of rare variants through the STRING PPI network.** In these stability paths plots, the x axis represents the steps taken along the lambda sequence to choose the optimal amount of regularisation required by the LASSO; the y axis represents the selection

probability of a gene. Selection probability paths (trajectories) for different genes are represented by different colours. Genes whose trajectories crossed the threshold of 0.80 selection probability were considered as robust predictors of case-control status and followed up in subsequent analyses. The selection of *PFAS* is here replicated.
(DOCX)

**S3 Fig. Stability selection on (A) the raw ("unsmoothed", α = 0) mutation profile in ADNI**; (B) the mutation profile in ADNI smoothed through a randomised version of the hippocampus network; (C) the mutation profile in ADNI smoothed through a non-brain- related network (umbilical cord). No genes were selected in any of these negative controls with probability higher than 80%.
(DOCX)

**S4 Fig. Stability selection on brain structures and components (other than the hippocampus)** from Greene et al.: top, ADNI WGS data smoothed through the neuron network; middle, ADNI WGS data smoothed through the medulla oblongata network; bottom, ADNI WGS data smoothed through the diencephalon network.
(DOCX)

**S5 Fig. Stability selection on a gene burden including only rare deleterious stop-gain, stop-loss and frameshift mutations, smoothed through** (A) the hippocampus network from Greene et al (2015) (13,616 genes tested); (B) the STRING PPI network (13,385 genes tested). No genes were selected with probability higher than 80% in either of these alternative scenarios.
(DOCX)

**S6 Fig. Randomisation control over the amount of overlap between NETPAGE- selected genes in ADSP and the interactome of PFAS.** Eighteen out of 29 genes selected in ADSP were observed in the interactome of PFAS (red vertical line). We formed 10,000 replicates of 29 genes selected randomly from the 16,298 genes tested, and counted how many genes in these replicates were also present in the interactome of PFAS. The amount of overlap observed with the genes resulting from the ADSP experiment could not have been achieved by chance, but suggests the presence of functional connections linking NETPAGE's results in the two independent datasets.
(DOCX)

**S7 Fig. Results of gene-based rare variant association testing in ADNI:** (A) SKAT-O and (B) mass-univariate testing of smoothed scores against case-control status, in 439 Caucasian participants. Both tests were performed correcting for sex, age, years of education, number of *APOE* 4 alleles, and first two principal components population substructure.
(DOCX)

**S8 Fig. Results of gene-based rare variant association testing (SKAT-O) in 10,186 unrelated individuals of Caucasian ancestry from ADSP (16,630 genes tested).** The gene-wide significance threshold (red line) was set at $0.05/16,630 = 3 \times 10^{-6}$. Gene-based models were corrected for sex, age, number of APOE ε4 alleles, and sequencing centre.
(DOCX)

**S9 Fig. Survival analysis for the gene resulting from stability selection in ADNI.** (A) Kaplan-Meier survival probability curves stratified by mutation status for *PFAS*. (B) The forest plot depicts hazard ratios from Cox proportional hazards model with confidence intervals and statistical significance. There was still no association between mutation status for PFAS and risk of conversion after covariate correction. (C) Results of the same Cox model fitting as in Fig 4, conducted after removing ADNI subjects diagnosed as cognitively normal at the latest

time point available. Therefore the only samples where the conversion event did not occur used in this model were individuals with a stable diagnosis of MCI at the latest time point available. This aimed at partially avoiding circular analysis issues, as cognitively normal individuals were used as control samples in the discovery phase involving stability selection (Fig 3). The score resulting from network propagation for *PFAS* was still seen to be significantly associated with lower risk of conversion to AD after restricting to stable MCI the samples where the conversion event did not occur.
(DOCX)

**S10 Fig. Reduction and visualisation in semantic space, through ReViGO, of the 928 GO terms significantly enriched (at $p_{FDR} < 0.05$) among the 1,449 first and second neighbours of PFAS in the hippocampus network.** Top, ontology terms from GO Biological Process; middle, terms from GO Cellular Component; bottom, terms from GO Molecular Function. Bubble color indicates the term p-value (colour bar in lower right-hand corner); size indicates the frequency of the GO term in the underlying database (bubbles of more general terms are larger).
(DOCX)

**S11 Fig. Distributions of enrichment p-values for the seven AD-related CGP gene sets, and 1000 gene sets randomly sampled from the background; the red line indicates the location of the non-randomised, uncorrected p-values (P column in S3 Table).** We show that the significant overlap seen between our genes of interest and the AD-related gene sets curated by Blalock (S3 Table) could not be achieved by chance, as none of the 1000 randomly drawn gene sets achieved smaller p-values.
(DOCX)

**S12 Fig. Differential expression analysis for 30 selected genes in the Mount Sinai Brain Bank parahippocampal gyrus expression dataset.** Full numerical results for all pairwise comparisons and their significance are provided in S4 Table.
(DOCX)

**S13 Fig. Randomisation control over the differential expression analysis in the Mount Sinai Brain Bank dataset.** We formed 1,000 sets of 30 randomly sampled genes and tested each of them for pairwise differences in expression levels among CERAD categories. We then counted how many genes in each set showed significantly different expression in at least one comparison and plotted their distribution. The red line indicates the 16 genes that showed significantly different expression in the original gene set of interest (i.e., the 30 genes selected in ADNI and ADSP). The data clearly shows that this amount of dysregulation could not be observed by chance, but is indeed linked to the overrepresentation of disease-related genes in the set of interest.
(DOCX)

## Acknowledgments

Some of the computing for this project was performed on the Sherlock 2.0 cluster. We would like to thank Stanford University and the Stanford Research Computing Center for providing computational resources and support that contributed to these research results. We would also like to thank Prof Jonathan M. Schott from the UCL Dementia Research Centre for his valuable clinical insights.

## Author Contributions

**Conceptualization:** Andre Altmann.

**Data curation:** Marzia Antonella Scelsi, Valerio Napolioni.

**Formal analysis:** Marzia Antonella Scelsi.

**Funding acquisition:** Michael D. Greicius, Andre Altmann.

**Investigation:** Marzia Antonella Scelsi, Andre Altmann.

**Methodology:** Marzia Antonella Scelsi, Andre Altmann.

**Resources:** Andre Altmann.

**Software:** Marzia Antonella Scelsi.

**Supervision:** Andre Altmann.

**Visualization:** Marzia Antonella Scelsi.

**Writing – original draft:** Marzia Antonella Scelsi.

**Writing – review & editing:** Marzia Antonella Scelsi, Valerio Napolioni, Michael D. Greicius, Andre Altmann.

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
