## [Decision Letter · Decision Letter 0]

3 Jun 2020

Dear Ms Scelsi,

Thank you very much for submitting your manuscript "Network propagation of rare variants in Alzheimer’s disease reveals tissue-specific hub genes and communities" for consideration at PLOS Computational Biology.

As with all papers reviewed by the journal, your manuscript was reviewed by members of the editorial board and by several independent reviewers. In light of the reviews (below this email), we would like to invite the resubmission of a significantly-revised version that takes into account the reviewers' comments.

Specifically, while the reviewers appreciate extension of network propagation methods for gene identification using a tissue-specific network and the biologically interesting results of your analysis, the raise several concerns. The most important issues to be addressed in a revision are (please see below for details and other comments): (i) a comparison to / distinction of your method from state of the art network propagation approaches; (ii) additional evidence for robustness of the analysis results and their biological interpretation (e.g., regarding gene selection criteria and effects of network topology).

We cannot make any decision about publication until we have seen the revised manuscript and your response to the reviewers' comments. Your revised manuscript is also likely to be sent to reviewers for further evaluation.

Sincerely,

Joerg Stelling

Associate Editor

PLOS Computational Biology

Arne Elofsson

Deputy Editor

PLOS Computational Biology

Reviewer's Responses to Questions

**Comments to the Authors:**

Reviewer #1: Scelsi et al. have presented a new program, NETPAGE, of gene prioritization and have applied it to detect relevant genes associated with Alzheimer disease (AD). The program uses network propagation, and in this sense, it is not novel. However, they have used a very accurate protocol to infer the likelihood of their predictions, using bootstrap resampling and sparse regression selection; also, they have used a novel tissue-specific network to study the effect of rare deleterious genes associated with AD. The approach is very interesting and produces remarkable results. Nevertheless, I would need some extra explanations to clarify some details of their approach.

There are some parameters, such as the mid-range α value of 0.5 and a top edge percentage of 1%, that need to be better explained, proving that they are optimal. For example, I would have expected the use of a 10 or 5 fold approach with known associated genes to optimise the prediction, taken the appropriate care on the use of a negative set (for example, I agree with some of the definitions used by the authors, randomising the network, although other approaches may also be valid).

Some relatively recent or well established works on networks have used other approaches, instead of selecting a fixed top percentage threshold. For example, the functional-based selection of top-ranking genes, which is a procedure to identify if the set of ranking genes is functionally similar to the set of initial seed genes. This procedure was implemented in Ghiassian et al. [1]in DIAMOnD. The authors should proof that the selection of 1% top-ranking genes produces acceptable results (only as an example, Sharma et al. showed that DIAMOnD outperformed existing algorithms in predicting asthma related genes[2], the authors may not need to proof their approach is better, but using a threshold of 1% should be at least compared with other values or criteria).

Finally, the authors have tested the robustness of their approach. Specifically, the sentence is: “Lastly, we investigated the robustness of the method with respect to the network structure and derivation methods by using the human, non- tissue-specific protein-protein interaction network (PPI) available through the STRING database”. I think there are some important issues for this analysis:

1) using the STRING database was an excellent approach when they decided to analyse a functional network, but it is not the right choice to analyse the protein interaction network. I would suggest using the recently published network of human the interactome by Luck et al. [3].

2) Some other works have tested the robustness of network propagation approaches for gene-prioritization [4] and the relevance of the network completion (i.e. using other near or proximal neighbouring search approaches [5]). Interestingly, not all diseases have a network connectivity that allows for the use of network propagation approaches to predict gene-disease associations or infer relevant pathways. Therefore, the robustness and network-features for this study should be more deeply studied, above all for its dependency on the network and on the network completion (i.e. analyses on the connectivity of gene-disease associations depending on the network, testing artificially the effect of edge removal or the dependency on the number of seeds used for the propagation).

Overall, I think the statistic approaches used in this work are really interesting and the results are coherent and important, even if the method itself (or its basic principle) is not new.

References

1. Ghiassian SD, Menche J, Barabasi AL. A DIseAse MOdule Detection (DIAMOnD) algorithm derived from a systematic analysis of connectivity patterns of disease proteins in the human interactome. PLoS Comput Biol. 2015;11(4):e1004120. doi: 10.1371/journal.pcbi.1004120. PubMed PMID: 25853560; PubMed Central PMCID: PMCPMC4390154.

2. Sharma A, Menche J, Huang CC, Ort T, Zhou X, Kitsak M, et al. A disease module in the interactome explains disease heterogeneity, drug response and captures novel pathways and genes in asthma. Hum Mol Genet. 2015;24(11):3005-20. Epub 2015/01/15. doi: 10.1093/hmg/ddv001. PubMed PMID: 25586491; PubMed Central PMCID: PMCPMC4447811.

3. Luck K, Kim D-K, Lambourne L, Spirohn K, Begg BE, Bian W, et al. A reference map of the human binary protein interactome. Nature. 2020. doi: 10.1038/s41586-020-2188-x.

4. Guney E, Oliva B. Analysis of the robustness of network-based disease-gene prioritization methods reveals redundancy in the human interactome and functional diversity of disease-genes. PLoS One. 2014;9(4):e94686. Epub 2014/04/16. doi: 10.1371/journal.pone.0094686

PONE-D-14-06874 [pii]. PubMed PMID: 24733074.

5. Menche J, Sharma A, Kitsak M, Ghiassian SD, Vidal M, Loscalzo J, et al. Disease networks. Uncovering disease-disease relationships through the incomplete interactome. Science. 2015;347(6224):1257601. doi: 10.1126/science.1257601. PubMed PMID: 25700523; PubMed Central PMCID: PMCPMC4435741.

Reviewer #2: The authors have provided a good approach for network propagation of rare variants, following a natural extension of the established literature. However, I have significant concerns, especially in the area of the strength of their network controls. Specifically, the authors have not convinced me that they are correctly controlling for the network topology; in particular, I would like to see their tests performed on null networks that approximately preserve the degree of all genes but scramble the identity of edges. See for instance:

Huang, J. K., Carlin, D. E., Yu, M. K., Zhang, W., Kreisberg, J. F., Tamayo, P., & Ideker, T. (2018). Systematic evaluation of molecular networks for discovery of disease genes. Cell systems, 6(4), 484-495.

Although you perform a single test against a randomized version (which should be degree-preserving, so that genes of high degree stay high degree, simple edge shuffling will destroy this relationship) of the network for Figure S3, there is not enough statistical power for me to be sure that this is well controlled. A test of 30 iterations or so (enough to get a variance estimate on the number of associated genes) should show that gene association is not solely influence by the degree of the genes that are selected, and would convince me that the real underlying network is contributing to detecting disease association. I would also like to see you address the underlying question of degree directly; is there a bias toward hubbier genes? Are hubby genes that have initial score more or less contributing to the final association than sparsely connected genes?

Along these lines, the ADSP set should be run on the alternative networks derived from other tissues. I see no indication that tissue specificity of the networks is contributing to better results. In the case of ADNI, you find a single gene association, PFAS, whereas running on the medulla oblongata net finds one as well, ZBBX. Is there any reason to indicate that the tissue specificity helps? Again, a series of randomized controls can help here, since the correct tissue specific network should outperform its corresponding null networks better than the incorrect tissue, but it needs a statistical argument.

Reviewer #3: The authors present NETPAGE, a method to investigate how rare variants spread in gene interaction networks. NETPAGE combines a network propagation approach with LASSO regression to find genes that are robustly high-scoring when changing a sparsity parameter. The authors apply NETPAGE to two Alzheimer's GWAS datasets, find high-scoring genes, and analyze the neighborhood of these genes. The paper is well written and clear, the results are interesting to read, and the authors do a good job of performing different complementary analyses on the NETPAGE results (e.g. survival analysis, gene set enrichment, RNA-seq overexpression). However, I have two related concerns.

First, NETPAGE is not a particularly novel method. There are many methods that do some type of network-based diffusion to find genes associated with a disease. The authors mention approaches such as CATAPULT, HotNet2, and NBS in the discussion but note that these are not appropriate to benchmark because they are fundamentally solving different problems. However, these approaches and others perform a similar type of analysis - for example, HotNet2 has been applied to complex diseases beyond cancer [Nakka 2016]. Further, [Lee 2011] models uncertainty of GWAS associations for network diffusion methods. Finally, [Lancour 2018] seems to have nearly the same goals as NETPAGE, using a network-diffusion approach to improve prediction of disease genes, and applied it on a large Alzheimer's GWAS dataset. These papers are the first of many that the authors need to either justify how their approach is substantially different or directly compare in the manuscript.

Second, the authors do not adequately justify that their method is finding biological signal within the datasets. One way to demonstrate this is through a classification-type framework, which they explicitly avoid since their problem is not one of a classification task. However, in that case, the subsequent simulations are even more important for their justification of the method. The simulations mutate some combination and percentage of first and second neighbors for a target protein - this needs to consider how we expect SNVs to be distributed through the network. Further, the simulations are run on subnetworks with first and second neighbors only, which effectively removes the potential signal from more distantly related genes. When the authors run NETPAGE on the two AD datasets, Fig S3 shows that these results are not recovered when using the non-smoothed scores, using a randomized brain network, or using another tissue network. However, are the best-performing genes the ones that are reported for the AD datasets on the original brain network? Why use the 80% cutoff? If the trends (e.g. relative order of important nodes) are consistent when randomizing the network or using non-smoothed scores, this could indicate that NETPAGE amplifies the signal but does not make new discoveries.

The application of NETPAGE to two different AD datasets reveals some intriguing overlaps related to PFAS and the neighborhood of PFAS genes. They compare to SKAT-O, but again it is unclear whether the NETPAGE results are biologically relevant so the comparison to SKAT-O is hard to interpret. They conduct a number of other experiments on the resultant genes and neighborhoods. First, the choice of focusing on neighborhoods is an intuitive one but not necessarily well-justified in terms of the results. Why not look at direct neighbors only? It is unclear what the outcome of the gene set enrichment analysis is: some of the significant gene sets are related to Alzheimer's, but the paper reports thousands of gene sets and it is unclear whether the AD-related gene sets would rise to the top of this list. Since PFAS is such an integral part of the results, the role of this gene and resulting mutations deserves more discussion.

There is a substantial section of the data availability in the Supplementary Methods. However, it is unclear what "Some restrictions will apply" means in the submission questions.

Minor Comments:

- Figure 2 should use "cases", "controls", "diffusion length" and "quantile normalization" as labels.

- In the Results under "Medium sample:ADSP", what is the "PFAS interactome"? I understand that PFAS is the predicted gene from the ADNI samples, but how this makes an interactome needs more clarification.

- Is there a reason why the authors considered binary variants rather than some p-value or risk score for SNVs?

REFS:

[Lee 2011] Lee et al., Prioritizing candidate disease genes by network-based boosting of genome-wide association data. Genome Research 2011.

[Nakka 2016] Nakka et al. Gene and Network Analysis of Common Variants Reveals Novel Associations in Multiple Complex Diseases. Genetics 2016.

[Lancour 2018] Lancour et al., One for all and all for One: Improving replication of genetic studies through network diffusion. PLoS Genetics 2018.

**Have all data underlying the figures and results presented in the manuscript been provided?**

Reviewer #1: Yes

Reviewer #2: Yes

Reviewer #3: Yes

PLOS authors have the option to publish the peer review history of their article (what does this mean?). If published, this will include your full peer review and any attached files.

Reviewer #1: No

Reviewer #2: No

Reviewer #3: No
---

## [Decision Letter · Decision Letter 1]

10 Nov 2020

Dear Ms Scelsi,

We are pleased to inform you that your manuscript 'Network propagation of rare variants in Alzheimer’s disease reveals tissue-specific hub genes and communities' has been provisionally accepted for publication in PLOS Computational Biology.

Best regards,

Arne Elofsson

Deputy Editor

PLOS Computational Biology

Arne Elofsson

Deputy Editor

PLOS Computational Biology

Reviewer's Responses to Questions

**Comments to the Authors:**

Reviewer #1: I thank the authors for their work in the revision, they have answered all my questions with excellence, I really appreciate.

Reviewer #2: This work is of interest to the field, but is rather technical in nature.

Reviewer #3: The authors addressed all my concerns and made clarifications where needed. Their new supplemental results (e.g., applying NETPAGE to the HuRI network and the randomized experiments) strengthen the paper and the authors' conclusions.

**Have all data underlying the figures and results presented in the manuscript been provided?**

Reviewer #1: Yes

Reviewer #2: Yes

Reviewer #3: Yes

PLOS authors have the option to publish the peer review history of their article (what does this mean?). If published, this will include your full peer review and any attached files.

Reviewer #1: No

Reviewer #2: **Yes: **Daniel Carlin

Reviewer #3: No

---

## [Editor Report · Acceptance letter]

31 Dec 2020

PCOMPBIOL-D-20-00470R1 

Network propagation of rare variants in Alzheimer’s disease reveals tissue-specific hub genes and communities

Dear Dr Scelsi,

I am pleased to inform you that your manuscript has been formally accepted for publication in PLOS Computational Biology. Your manuscript is now with our production department and you will be notified of the publication date in due course.

With kind regards,

Livia Horvath
